# Consecutive Injection of High-Dose Lipopolysaccharide Modulates Microglia Polarization via TREM2 to Alter Status of Septic Mice

**DOI:** 10.3390/brainsci13010126

**Published:** 2023-01-11

**Authors:** Zhiyun Qiu, Huilin Wang, Mengdi Qu, Shuainan Zhu, Hao Zhang, Qingwu Liao, Changhong Miao

**Affiliations:** 1Department of Anesthesiology, Zhongshan Hospital, Fudan University, Shanghai 210000, China; 2Shanghai Key Laboratory of Perioperative Stress and Protection, Shanghai 210000, China

**Keywords:** sepsis, microglia polarization, neuroinflammation, TREM2

## Abstract

Background: The neuroinflammation of the central nervous system (CNS) is a prevalent syndrome of brain dysfunction secondary to severe sepsis and is regulated by microglia. Triggering the receptor expressed on myeloid cells 2 (TREM2) is known to have protective functions that modulate the microglial polarization of M2 type to reduce inflammatory responses, thereby improving cognition. Methods: We examined the effect of TREM2 on the polarization state of microglia during the progression of neuroinflammation. After consecutive intraperitoneal injections of lipopolysaccharide for 7 days, we evaluated the inflammation of a septic mice model by hematoxylin–eosin (H&E) and electron microscopy, and we used immunofluorescence (IF) assays and Western blotting to visualize hippocampal sections in C57BL/6 mice to assess TREM2 expression. In addition, we analyzed the state of microglia polarization with quantitative RT-PCR. Result: The consecutive injection of LPS for 4 days elevated systemic inflammation and caused behavioral cognitive dysfunction in the septic model. However, on Day 7, the neuroinflammation was considerably attenuated. Meanwhile, TREM2 decreased on Day 4 and increased on Day 7 in vivo. Consistently, LPS could reduce the expression of TREM2 while IFN-β enhanced TREM2 expression in vitro. TREM2 regulated the microglial M1 phenotype’s conversion to the M2 phenotype. Conclusion: Our aim in this study was to investigate the interconnection between microglia polarization and TREM2 in neuroinflammation. Our results suggested that IFN-β could modulate TREM2 expression to alter the polarization state of microglia, thereby reducing LPS-induced neuroinflammation. Therefore, TREM2 is a novel potential therapeutic target for neuroinflammation.

## 1. Introduction

Sepsis, an organ dysfunction syndrome with abnormal physiological responses caused by infection, is a major cause of death and critical illness worldwide [1]. As one of the vital organs damaged by sepsis, the central nervous system further contributes to the development of neuroinflammation due to the systemic inflammatory response that activates microglia cells in the brain to disrupt the blood–brain barrier [2] and the release of pro-inflammatory and chemotactic factors by activated microglia cells [3,4,5]. Microglia, as immune cells in the central system, have two classical states: M1 (pro-inflammatory) and M2 (anti-inflammatory). M1-phenotype microglia promote the secretion of inflammatory cytokines and increase the expression of markers, such as CD86, INOS, and CD16, which induce an inflammatory response in the brain, while M2-phenotype microglia express anti-inflammatory cytokines and markers, such as CD206, transforming cytokines Arg-1 and IL-10. Although this categorization might be an oversimplification, classifying microglia functions as either neurotoxic (M1) or neuroprotective (M2) is useful for explaining inflammatory pathobiology. Therefore, we will adopt the binary classification of microglia to discuss microglia polarization from M1 to M2. In the presence of neuroinflammation, a type I interferon (IFN-I) sometimes plays a neuroprotective role in central neurological diseases, such as multiple sclerosis (MS) and experimental autoimmune encephalomyelitis (EAE) [6,7,8]. IFN-β is a type I IFN induced in APCs by TLR stimuli, which is used for the treatment of MS. Its mechanisms of action include the induction of IL-10, inhibition of the expansion of encephalitogenic T cells, reduced production of inflammatory cytokines, and a reduction in blood–brain barrier permeability [9,10]. A lack of IFN-β leads to the persistent activation of residual APCs in CNS, which results in prolonged inflammation and extensive demyelination. Thus, IFN-β is used clinically as one of the drugs to alleviate the neuroinflammatory response [11]. It can also reduce microglial responsiveness, microglial proliferation, and neuroinflammation [12].

Triggering receptor expressed on myeloid cells 2 (TREM2) is a considerable innate immune receptor that is primarily expressed on microglia in CNS. Furthermore, TREM2 plays an important role in regulating the microglial functions that lead to disease progression [13]. It suppresses the production of pro-inflammatory factors by binding to the DNAX-activating protein of 12 kDa (DAP12) to regulate the inflammatory response [14]. Moreover, TREM2 also has phagocytosis, a function that has been shown to be instrumental in many degenerative diseases associated with neuroinflammation, such as Alzheimer’s disease (AD), MS, Parkinson’s disease (PD), etc. A number of studies have reported that the expression of TREM2 is significantly decreased in LPS-induced sepsis models [15,16,17]. TREM2, expressed in macrophages, affects inflammation through Toll-like receptor (TLR)-dependent pathways [18]. Furthermore, TREM2 negatively regulates TLR-induced responses through an unknown mechanism [16]. Mice TREM2 knockdown produces more inflammatory cytokines in response to LPS. In contrast, TREM2 knockdown inhibits microglia activation [19] and increases the expression of pro-inflammatory genes. Nevertheless, the mechanisms regulating IFN-β and TREM2 in neuroprotection are unknown.

In our study, we reported that IFN-β regulated TREM2 expression and triggered the conversion of M2 (anti-inflammatory) phenotype-microglia to neuroinflammation.

## 2. Materials and Methods

### 2.1. Animals and Cell Line

We obtained male wide-type C57BL/6 (WT; 6–9 weeks) mice from Shanghai Jihui Laboratory Animal Care (Shanghai, China). We maintained mice under a 12 h light/dark cycle with food and water available ad libitum. All procedures were approved by the Animal Care and Use Committee of Zhongshan Hospital, Fudan University (Shanghai, China) (approval code 202108006s). We performed all experiments in accordance with guidelines from the Chinese Animal Welfare Agency.

We maintained the murine microglial cell line BV-2 in culture media RPMI-DMEM (Gibco) supplemented with 10% fetal bovine serum (FBS; Sigma Aldrich, Shanghai, China) and 1% penicillin–streptomycin solution at 37 °C and 5% CO_2_ in a humidified incubator.

### 2.2. Murine Sepsis Score Test

The murine sepsis score (MSS) included variables affected by infection, including spontaneous activity, response to touch and auditory stimuli, posture, respiration rate and quality (labored breathing or gasping), and appearance (the degree of piloerection). We provided a score between 0 and 4 for each of these variables.

### 2.3. Cell Culture and Transfection

We routinely cultured mouse microglia BV-2 (Derived from C57BL/6 mouse microglia) cells in Dulbecco’s modified Eagle’s medium (DMEM) supplemented with 10% fetal bovine serum (FBS) and penicillin/streptomycin (PS). We treated cells with 1 ug/mL LPS for 1 h before treating with IFN-β for another 24 h.

### 2.4. Lentiviral Particles Preparation

We cloned a TREM2 short hairpin RNA sequence tagged with a GFP gene into the PFMLV-SC5 lentiviral vector to knockdown TREM2. Additionally, we inserted a scramble TREM2 short hairpin RNA sequence for the control study and GFP gene into the lentiviral vector. We purified lentiviral vectors and then transfected them into 293 T cells. We collected lentiviral particles after 48 h. We seeded BV-2 microglia into 6-well plates. We added lentiviral particles to the culture for 8–12 h. The efficiency of microglia transduction was at least 90%, as determined by the number of microglia expressing GFP fluorescence. After transfection, we treated cells with 1 ug/mL LPS and IFN-β for 24 h and 24 h, respectively.

### 2.5. Treatments of Mice Models

For LPS-induced inflammation, we injected mice intraperitoneally with 8 mg/kg LPS derived from *E. coli* serotype 055: B5 (Sigma, L2880) or an equivalent volume of drug (PBS) daily for 4 or 7 days. We identically treated six mice per group. A total of 24 h after the last LPS injection, we euthanized the animals then perfused and harvested the brains.

Then, we collected 200–300 µL of blood from the eyes. We kept tubes containing blood samples at 4 °C for 24 h and then centrifuged at 7500× *g* for 10 min. We collected clear supernatant. Then, we diluted the serum and used it for ELISA for cytokine measurement.

### 2.6. Novel Object Recognition (NOR)

After 10 min of habituation in the open field in the three previous days, we allowed the animals to explore two identical objects (ball-shaped objects) in an open field (33 cm × 33 cm × 40 cm) for five minutes (trial-session-encoding phase). After 1 h in the home cage (consolidation phase), we conducted the retention trial (testing-session-retrieval phase) and replaced one of the previously presented objects with a novel object (transparent glass cylinders). During the 10 min test, we measured the duration of exploration of each object. We performed the experiment in an isolated room under direct overhead lighting. We calculated the NOR discrimination index according to the following formula: time of novel object exploration/(time of novel + familiar object exploration) × 100. We immediately sacrificed the mice exposed to the cognitive test at the end of the encoding or retention phase.

### 2.7. Y-Maze

We created a Y-Maze with three arms (20 cm long × 10 cm wide × 20 cm high) at 120° angles, designated A, B, and C. We placed the mice in the distal end of arm A and allowed them to explore the maze for 10 min. A video camera mounted above the maze recorded the movements of the mice for analysis. We recorded the arm entries and calculated the percentage of alternations (entry into an arm that differs from the previous two entries) with the following formula: Alternations/(ArmEntries − 2) × 100.

### 2.8. Hematoxylin and Eosin (H&E) Staining

We fixed brain tissues in 4% paraformaldehyde for 24 h and 5 μm thick sections of coronal.

We cut hippocampal sections on the slides, deparaffinized, and then stained with hematoxylin and eosin. We examined the sections under a light microscope (Leica, Weztlar, Germany).

### 2.9. Immunofluorescence

We rapidly excised brain tissues, rinsed them with PBS, infused them in 4% buffered paraformaldehyde, and embedded them. We sectioned the brain tissues at 5 μm thickness in paraffin, before deparaffinizing in xylene and rehydrating in ethanol. We blocked the slides using 1% BSA. Then, we incubated with primary antibodies Iba-1 and TREM2 (Abcam) with a dilution of 1:100 at 4 °C overnight, respectively, followed by staining with secondary antibody at 1:200 for 1 h at room temperature. We washed the slides three times with PBS. Then, we covered the sections with glass using mounting solution and examined using laser scanning confocal microscopy (Leica, Germany).

### 2.10. Western Blot

We treated BV-2 cells or murine hippocampal tissues from different groups by RIPA lysate buffer (Solarbio, R0020) to extract protein samples. Hereinto, murine brain tissues first underwent grind prior to the addition of RIPA lysate buffer. We loaded protein samples, separated on 10% SDS gels, and transferred onto a polyvinylidene fluoride (PVDF) membrane. We placed the membrane into 5% milk blocking solution for 1 h at room temperature, incubated with primary antibodies overnight at 4 °C, and incubated with secondary antibodies after washing with TBST. We captured the chemiluminescence signals of protein bands using Tanon 5200 Chemiluminescent Imaging System (Tanon, 5200). The antibodies used in this experiment were as follows: TREM2 (1:1000 dilution, Cell Signaling Technology, 76765), GAPDH (1:5000 dilution, Zen-BIO, 200306−7E4), HRP-conjugated goat anti-mouse IgG (1:5000 dilution, Beyotime, A0216), and HRP-conjugated goat anti-rabbit IgG (1:5000 dilution, Beyotime, A0208).

### 2.11. ELISA Assay

We collected cell supernatants or murine Hippocampal homogenates from different groups at the indicated time points. We used ELISA kits purchased from Shanghai, China. to detect the contents of IL-10 (abs520005), IFN-β, IL-6 (abs520004), and TNF-α (abs520010). All procedures were in accordance with the manufacturer’s instructions. We measured the absorbance at a wavelength of 450 nm on a microplate reader (ThermoFisher, Multiskan^TM^, 51,119,000).

### 2.12. Quantitative RT-PCR

We performed quantitative real-time PCR (qRT-PCR) using 4 μg of the total RNA as a template for the reverse transcription reaction using a TaKaRa 1st-strand kit (Dalian, China). We performed reactions according to the manufacturer’s protocol. We subjected synthesized cDNA to real-time PCR assays with specific primers and SYBR@Premix Ex TaqTM qPCR SuperMix (Takara, Dalian, China). We used GAPDH as the internal control and applied the 2^−ΔΔCt^ method to calculate the experimental groups’ fold-change differences compared with the control group. The qPCR primers are listed in Table 1. We normalized the final results, which we expressed as the fold change compared with the target gene/GAPDH.

### 2.13. Statistical Analysis

We performed statistical analysis using GraphPad Prism (v.6–8; Graph-Pad) and Microsoft Excel (2010 and 2016; Microsoft). Unless indicated otherwise, data is represented as mean ± S.E.M. (standard error of the mean). The figures show the statistical significance. The statistical tests that underlie the data analysis are stated in the corresponding figure legends. For comparing the two datasets, we determined statistical significance by unpaired two-tailed *t*-test in the case of Gaussian and Mann–Whitney test in the case of non-Gaussian distribution. For comparing more than two datasets, we determined statistical significance with a one-way ANOVA with Tukey’s multiple comparisons test in the case of Gaussian and Kruskal–Wallis’ test with Dunn’s multiple comparisons test in the case of non-Gaussian distribution. In the case of the number of animals per experiment not being sufficient to test for normality, we assumed Gaussian distribution unless indicated. We set significance levels as * *p* < 0.05, ** *p* < 0.01, and *** *p* < 0.001.

## 3. Results

### 3.1. Reduction of TREM2 in Septic Model In Vivo and In Vitro

TREM2’s neuroprotective effects in neuroinflammation have been well documented [20]. TREM2 exerts anti-inflammatory effects and promotes apoptotic neurons in many diseases, including neurodegenerative diseases, ischemia/reperfusion injury, and bacterial infections phagocytosis [20,21,22,23,24]. Here, we studied TREM2 expression by Western blotting in mice brain tissues derived from LPS-induced mice or untreated controls. To quantify TREM2 expression, we performed quantitative real-time RT-PCR (RT-qPCR) studies, which demonstrated that TREM2 expression is significantly lower in LPS-induced mice compared with hippocampal tissues from controls before 4 days; however, it increased after the fourth day (Figure 1a,b). Our confocal analysis showed TREM2-specific expression in ionized calcium-binding adapter molecule 1 (Iba1) cells in active microglia. Our results suggested that TREM2 was significantly upregulated and overlapped with activated microglia Iba-1 in 7D mice (Figure 1c). Our results also implied that LPS-induced neuroinflammatory responses inhibited the expression of TREM2. However, we detected a considerable increase in TREM2 expression in 7D mice.

Furthermore, we examined CD86, iNOS, IL-10, and CD206 expression and found that there were distinctive changes in the brains of LPS-induced septic mice. We found that the percentage of CD86+ or iNOS+ microglia considerably increased in the brains of 4D mice but not in 7D mice. Moreover, the percentage of CD206+ or IL-10+ microglia considerably decreased in the brains of 4D mice but increased in 7D mice (Figure 1d). Our results suggested that pro-inflammatory microglia activated, while the number of anti-inflammatory microglia decreased in the brains of 4d mice. Therefore, TREM2 participated in the conversion of microglia to the M2 phenotype to attenuate neuroinflammatory response.

To further confirm our results in vivo, we also observed the intracellular expression of TREM2 in BV-2 cells. Our results showed that BV-2 cells treated with 1 μg/mL, 5 ug/mL, or 10 μg/mL LPS for 24 h had much lower expressions of TREM2 compared with the control group (Figure 2a–c). Among them, 1 ug/mL LPS treatment downregulated TREM2 levels to half of the control group. As is well understood, LPS promoted microglia polarization towards the M1 phenotype. At the same time, the M1 phenotype increased with increased concentrations of LPS treatment, reducing the M2 anti-inflammatory phenotype, which exacerbated the inflammatory response (Figure 2d). Our results indicated that LPS treatment could reduce TREM2 expression both in vitro and in vivo and alternate the microglial polarization status. To further verify whether LPS changed the polarization state of microglia via TREM2, we overexpressed TREM2 in BV-2. We found TREM2 to be considerably elevated in vitro by exogenous treatment (Appendix A); moreover, the same LPS treatment in BV-2 cells with overexpressed TREM2 revealed a decrease in M1-phenotype polarization from microglia (Appendix A).

### 3.2. Evaluation of Inflammation in Septic Mouse Model

Next, we detected the weight, blood neutrophil-to-lymphocyte ratio (NLR), and murine sepsis score in normal mice and mice undergoing consecutive injections of LPS. Our results showed that the weight, NLR, and murine sepsis score were hardly affected by 8 mg/kg of LPS for different consecutive injection times (ranging from 1 to 7 days) in mice (Figure 3a–c). The weight, NLR, and murine sepsis score were gradually lowered when the consecutive injection times of LPS were less than 4 days; however, it considerably increased when the consecutive injection times of LPS were more than 4 days. Although the appetite and weight decreased in the mice consecutively injected with LPS, there was no statistically significant difference in mortality among the three groups (Appendix A). In particular, the levels of pro-inflammation cytokines (TNF-α, IL-6) reached a peak value at the consecutive injection of LPS for 4 days; however, it dropped at the seventh day of consecutive LPS injection. By contrast, the level of anti-inflammation cytokine IL-10 was considerably increased at the seventh day of consecutive LPS injection (Figure 3d,e). Our results indicated that LPS-induced inflammation may reduce at the seventh day of consecutive LPS injection.

To verify whether the intraperitoneal injection of LPS could cause neuroinflammation, we conducted novel object recognition (NOR) and Y-maze experiments to explore whether learning and spatial memory abilities were impaired in mice that received injections over a long period of time. We performed behavioral tests on mice in the control group and each of groups injected with LPS on days one to seven, and we found a similar trend of post-impairment re-recovery of cognitive dysfunction in mice (Appendix A). In the absence of statistical differences in the total time spent exploring objects by the three groups of mice (Figure 4b), the novel object discrimination index (NODI) was significantly lower in the group of mice that received consecutive injections of LPS for 4 days (we use “4D” to represent this group of mice) compared with the vehicle and the group of mice that received consecutive injections of LPS for 7 days (we use “7D” to represent this group of mice) (Figure 4a). Similarly, in the Y-maze experiment, our data showed that 7D had more spontaneous alternation than 4D, which was not significantly different from the vehicle mice (Figure 4c). Thus, mice injected with LPS for 4 consecutive days had impaired cognitive function while mice injected with LPS for 7 consecutive days had restored cognitive function. By H&E staining of hippocampal tissue, we observed that the nerve cells in the hippocampus of 4D showed nucleus sequestration, deep staining, and morphological changes compared with the vehicle group (Figure 4d). Furthermore, the mitochondria of 4D by electron microscopy revealed that the mitochondrial membrane was broken while the mitochondrial cristae disappeared and we observed vacuole formation (Figure 4e), indicating that the mitochondrial structure destruction of mouse hippocampus in 4D would further cause the dysfunction of mitochondria.

We quantified the inflammatory cytokines’ expression of pro-inflammatory cytokines in the hippocampal tissue of LPS-induced mice. We found significant differences in the expression among the three different groups. Our results indicated increased expressions of TNF-α and IL-6 in 4D mice (Figure 5a,b); however, expression decreased in hippocampal tissues of 7D mice compared with the other two groups (Figure 5c,d). Next, we also quantified the mRNA expression of pro-inflammatory cytokines in the hippocampal tissue of LPS-induced mice. We found significant differences in the expression of IFN-β, IL-10, TNF-α, and IL-1β among the three different groups. While the mRNA expression of the inflammatory cytokine TNF-α and IL-1β significantly increased in the hippocampi of 4D mice compared with control mice and 7D, this change was reversed in 7D mice (Figure 5e,f). The data indicated that not only was the systemic inflammatory response activated in 4D mice, but it also affected the inflammatory response in the hippocampal tissue of LPS-induced mice. In 7D mice, the systemic inflammatory response was reduced; however, neuroinflammation was also restored.

### 3.3. IFN-β Enhanced TREM2 Expression Triggering Conversion of Microglia from M1 to M2

Because IFN-β and IL-10 levels in the hippocampal tissue were validated by ELISA and qPCR, we investigated whether there was a connection between IFN-β or IL-10 and TREM2. We handled BV-2 cells with different concentrations of IFN-β or IL-10 for 24 h in vitro. We found that TREM2 expression was improved after IFN-β treatment compared with the control (Figure 6a,b). Nevertheless, TREM2 expression was not significantly changed after treatment by IL-10 (Figure 6c,d). In particular, IFN-β treatment induced a concentration-dependent upregulation of TREM2 (Figure 6a,b,e). IFN-I has different roles in multiple diseases; however, in many neurological diseases, IFN-I is able to block pro-inflammatory mediators and induce the effects of anti-inflammatory factors [25].

TREM2 is able to affect Arg-1 expression, which is induced by anti-inflammatory factors, after affecting the expression of microglial transcription factor STAT6 [13]. This indicates that TREM2 plays an instrumental role in microglial polarization to the M2 phenotype and in promoting central nervous anti-inflammatory effects. To mimic the environment in vivo, we co-incubated 1 ug/mL of LPS for 1 h and then treated BV-2 cells with different concentrations of IFN-β for 24 h. We speculated that a certain concentration range of IFN-β has a promoting effect on the conversion of microglia to the M2 phenotype. We examined IL-10 and Arg-1 levels and revealed the activation of M2-phenotype microglia. We discovered that IL-10 and Arg-1 expression was significantly upregulated, while CD 86 and CD16 production decreased (Figure 6f). Therefore, our results demonstrated that a certain concentration range of IFN-β promoted microglia polarization to the M2 phenotype by increasing TREM2 expression, which was able to mitigate neuroinflammation.

### 3.4. Knockdown of TREM2 in Microglia Inhibited Conversion of Microglia from M1 Phenotype to M2 Phenotype

Next, we investigated the effect of TREM2 knockdown on the alternative anti-inflammatory activation of microglia in response to IFN-β treatment. We verified the expression of TREM2 after TREM2 knockdown and showed a significant decrease in TREM2 expression in sh-TREM2 compared with the NC group (Figure 7a,b). We observed the highest increase in TREM2 expression in the IFN-β group at 1 ug/mL. Hence, we set the intervention time and dose of IFN-β in the mice model as 1 ug/mL for 24 h. It was evident that the TREM2 protein levels in the sh-TREM2 group decreased after IFN-β treatment compared with the NC group (Figure 7c). When we examined the activation of the M1 and M2 phenotypes in microglia, we detected decreased M2-phenotype IL-10 and ARG-1 mRNA levels (Figure 7d). Our results indicated that TREM2 knockdown reduced the anti-inflammatory capacity of microglia, and exogenous IFN-β is able to recover a few portions of the TREM2 expression but cannot totally promote microglia to M2-phenotype conversion, which played a pivotal role in regulating neuroinflammation.

## 4. Discussion

We systematically investigated the role of TREM2 in CNS inflammation using a consecutive intraperitoneal injection of the LPS model. Our results indicated that there was an auto-anti-inflammatory mechanism in the brain of mice receiving consecutive LPS injections, which allowed them to resist exogenous endotoxin infection. Mice that received consecutive injections of LPS for 7 days regained body weight and had a reduced systemic inflammatory response compared with 4D mice. In CNS, 7D mice showed the reduced expressions of pro-inflammatory cytokines and increased ratios of M2-phenotype microglia. The Y-maze and NOR tests showed that 7D mice recovered learning and spatial memory functions. H&E staining and mitochondrial EM analysis of brain tissue demonstrated reduced histopathological damage in the hippocampus tissue of 7D mice. These results implicated improved behavioral and pathophysiological symptoms in septic mice receiving long-term intraperitoneal LPS injections.

Moreover, we found that mice that received consecutive injections of LPS for 4 days increased the polarization of microglia to the M1 phenotype, enhancing the inflammatory response of neuroinflammation. However, in 7D mouse brains, there was a considerable increase in the microglial M2 phenotype, accompanied with the increased expression of TREM2, indicating a reduced inflammatory response in the CNS environment.

Many studies in the field of neurodegenerative diseases have explored the vital role of TREM2 [13,20,26]. Although most studies focused on the phagocytosis of TREM2, the link between TREM2 and microglia polarization is also supported by the literature. The overexpression of TREM2 increases the number of ARG-1-positive microglia and the mRNA levels of Arg-1 and YM1/2, thereby reducing neuroinflammation in the hippocampal region, which is associated with a long-term high-fat diet [27]. This conclusion can be verified by our sepsis model. In 7D mice, TREM2 expression in the hippocampal tissue was increased, and the inflammatory response was mild. Notably, TREM2 was able to promote the expression of M2 markers, such as Arg-1, and promote the polarization of microglia towards the M2 phenotype [13]. Promisingly, one study confirmed that LPS can stimulate a decreased expression of TREM2 [28]; however, the connection between them needs to be further illustrated. In our study, we found that the expression of IFN-β and IL-10 was altered after LPS treatment. Further experiments revealed that only BV-2 treated with IFN-β resulted in altered TREM2 expression while IL-10 did not cause changes in TREM2. Therefore, we validated the relationship between IFN-β and TREM2 in BV-2 cells in vitro. TREM2 increases in IFN-β treated cells accelerated with increasing IFN-β levels over a range of concentrations. IFN-β treatment decreased CD86 and CD16 mRNA levels and increased Arg-1, CD206, and IL-10 levels, suggesting that IFN-β could promote the alternation of the M1 to the M2 phenotype in microglia. Above all, IFN-β is able to enhance TREM2 microglia levels. However, there are two major limitations in our study that could be addressed in future research. First, our study focused on TREM2 with 28 kd, which is now gradually represented in the unglycosylated form. Although widely accepted, it suffers from some limitations compared with the 44 kd glycosylated form. Second, this research focuses on one region of brain tissue. However, LPS injection can cause inflammatory responses in various parts of the brain, including the cerebellum, substantia nigra region, and cerebral cortex. Based on these two points, we will further investigate the role of TREM2 in other brain regions, as well as the function of the glycosylated fragments of TREM2.

Over the past few years, neuroinflammation has been confirmed to be strongly associated with a variety of neurodegenerative diseases [29,30]. The prolonged inflammatory environment exacerbated microglia depletion, and microglia’s loss of immune effects further promoted the pathological progression of degenerative diseases. Therefore, the treatment of many clinical neurological diseases requires the suppression of persistent inflammation as a therapeutic tool. In addition to promoting anti-inflammatory effects, TREM2 enhances the phagocytic capacity of microglia and plays a role in a variety of neurodegenerative diseases, such as AD, PD, and MS [31,32].

In our study, we are the first to show that the hippocampal expression of TREM2 alleviates the status of mice after consecutive LPS injections. The protective effect of TREM2 is likely attributable to the promotion of microglial M2 polarization and the suppression of neuroinflammation. Our study explored the role of TREM2 in sepsis caused by consecutive LPS injections and suggests that TREM2 might be a novel target for the intervention of neuroinflammation. Future studies are needed to confirm our findings in other appropriate sepsis or neurodegeneration models, which may provide deeper insight into these preliminary findings.

## 5. Conclusions

In summary, we systematically investigated the inflammatory role of TREM2 in the central nervous system after sepsis using high concentrations of consecutive LPS injections to establish a mouse sepsis model. Our results indicate that consecutive LPS injections attenuate the inflammatory response to sepsis by releasing IFN-β, resulting in the upregulation of TREM2 expression, which mediates the polarization of microglia from the M1 to M2 phenotype, reducing neuroinflammation. Our results show that TREM2 has a potential application in neuroinflammation immunotherapy.

## Figures and Tables

**Figure 1 brainsci-13-00126-f001:**
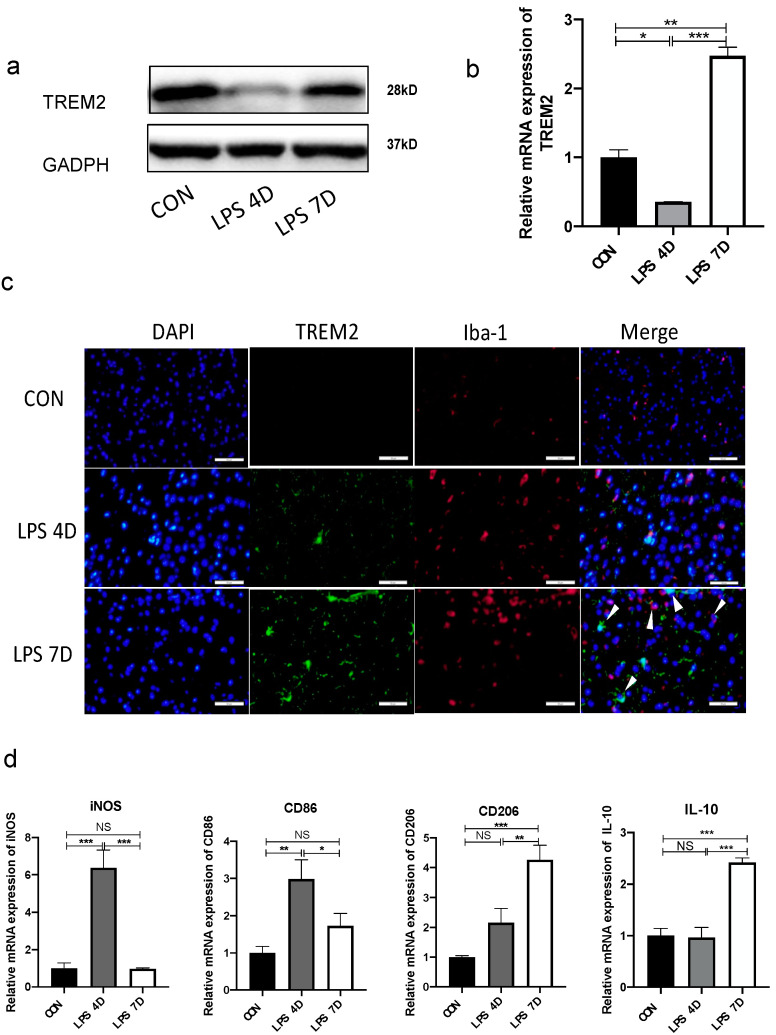
LPS reduced TREM2 expression and changed the polarization of microglia in vivo. (**a**) Protein levels of TREM2 showed opposite trends of these groups by Western blot. (**b**) mRNA expression of TREM2 in each group of mice by PCR. (**c**) Immunofluorescence staining of brain sections from each group of mice. Iba-1 indicates activated microglia (Red). (**d**) mRNA expression of M1-type microglia markers (iNOS and CD86) and M2-type microglia markers (CD206 and IL-10) by PCR. NS indicates no statistical difference from the control group. * *p* < 0.05, ** *p* < 0.03, *** *p* < 0.01 compared with control, n = 6 mice per group.

**Figure 2 brainsci-13-00126-f002:**
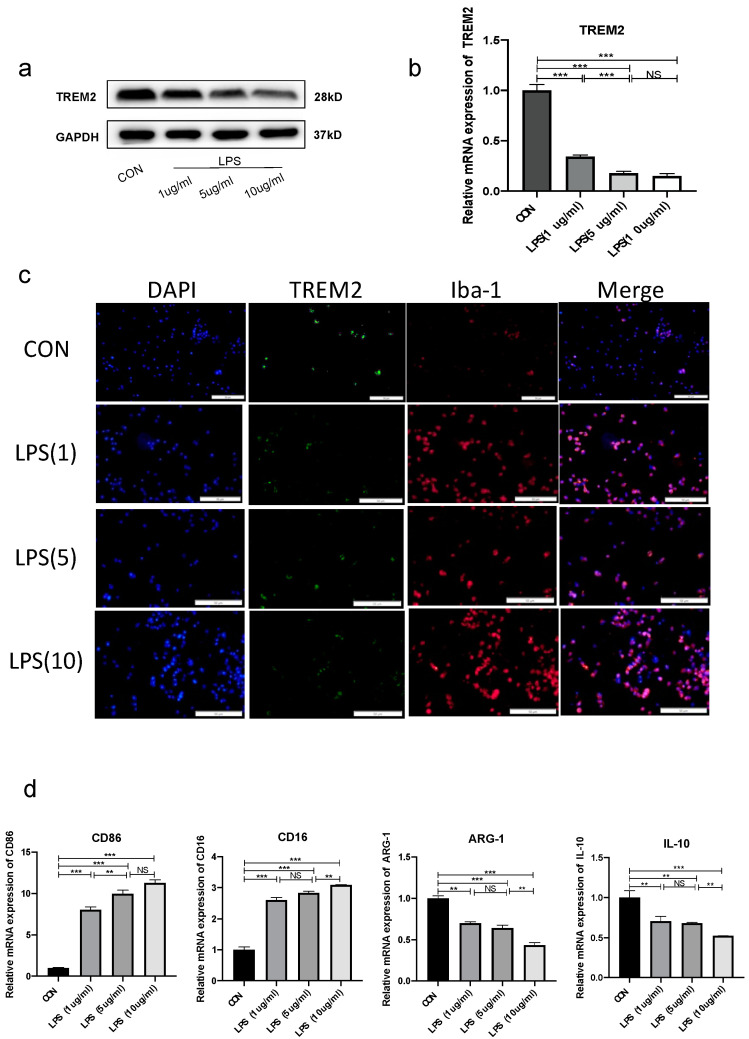
LPS reduced TREM2 expression and changed polarization of microglia in vitro. (**a**) Protein expression of TREM2 after 24 h treatment of BV-2 cells with different concentrations of LPS. (**b**) mRNA expression of TREM2 after 24 h treatment of BV-2 cells with different concentrations of LPS. (**c**) Immunofluorescence staining of BV-2 cells with different concentrations of LPS. (**d**) mRNA expression of M1-type microglia markers (CD16 and CD86) and M2-type microglia markers (ARG-1 and IL-10) by PCR. NS indicates no statistical difference from the control group. ** *p* < 0.03, *** *p* < 0.01 compared with control, n = 6 mice per group.

**Figure 3 brainsci-13-00126-f003:**
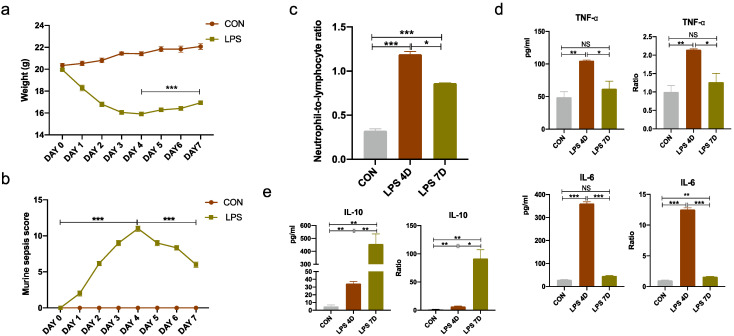
Reduced systemic inflammation in mice injected with LPS for 7 consecutive days. (**a**) Changes in body weight of mice. (**b**) Effects of LPS administration on murine sepsis score. (**c**) Ratio of neutrophils to lymphocytes in serum of each group of mice. (**d**) Expression of inflammatory factors TNF-α and IL-6 in mice serum measured by ELISA. (**e**) Expression of inflammatory factor IL-10 in mice serum measured by ELISA. NS indicates no statistical difference from the control group. * *p* < 0.05, ** *p* < 0.03, *** *p* < 0.01 compared with control, n = 6 mice per group.

**Figure 4 brainsci-13-00126-f004:**
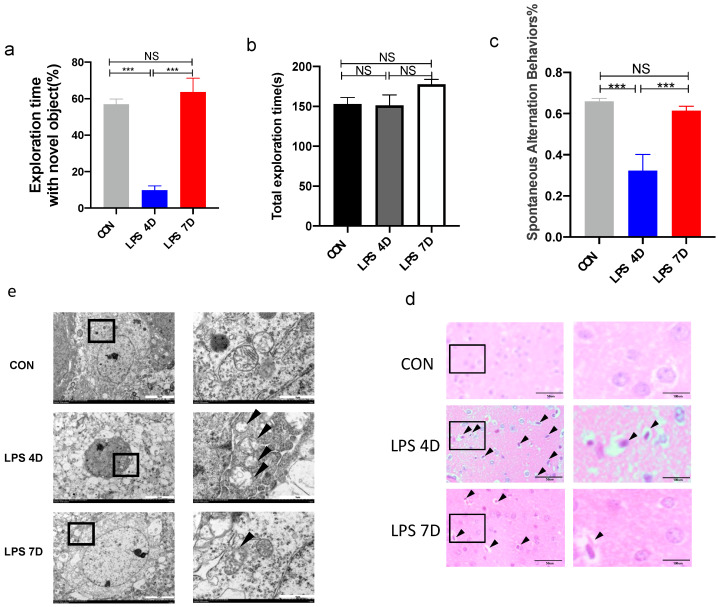
Improved cognitive dysfunction of mice injected for 7 consecutive days. (**a**) Percentage of detection time of new objects in each group of mice. (**b**) Total detection time of new and old objects for each group of mice. (**c**) Percentage of mice in each group entering the different arms of Y-maze in sequence. (**d**) HE staining of brain tissue sections of each group of mice. (**e**) Changes in mitochondria in central nervous system of mice in each group. NS indicates no statistical difference from the control group. *** *p* < 0.01 compared with control, n = 6 mice per group.

**Figure 5 brainsci-13-00126-f005:**
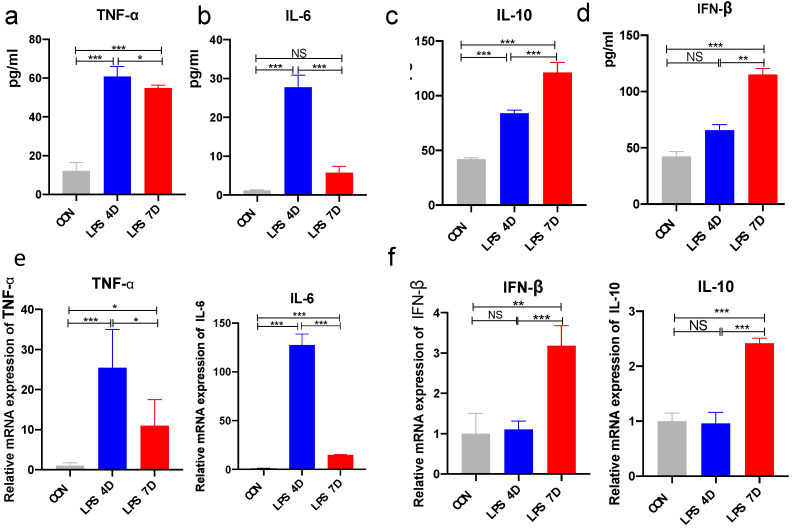
Neuroinflammatory response in hippocampal tissue of mice. (**a**–**d**) Expression of inflammatory factors TNF-α, IL-6, IL-4, and IL-10 in mice brains measured by ELISA. (**e**,**f**) mRNA expression of TNF-α, IL-6, IL-4, and IL-10. NS indicates no statistical difference from the control group. * *p* < 0.05, ** *p* < 0.03, *** *p* < 0.01 compared with control, n = 6 mice per group.

**Figure 6 brainsci-13-00126-f006:**
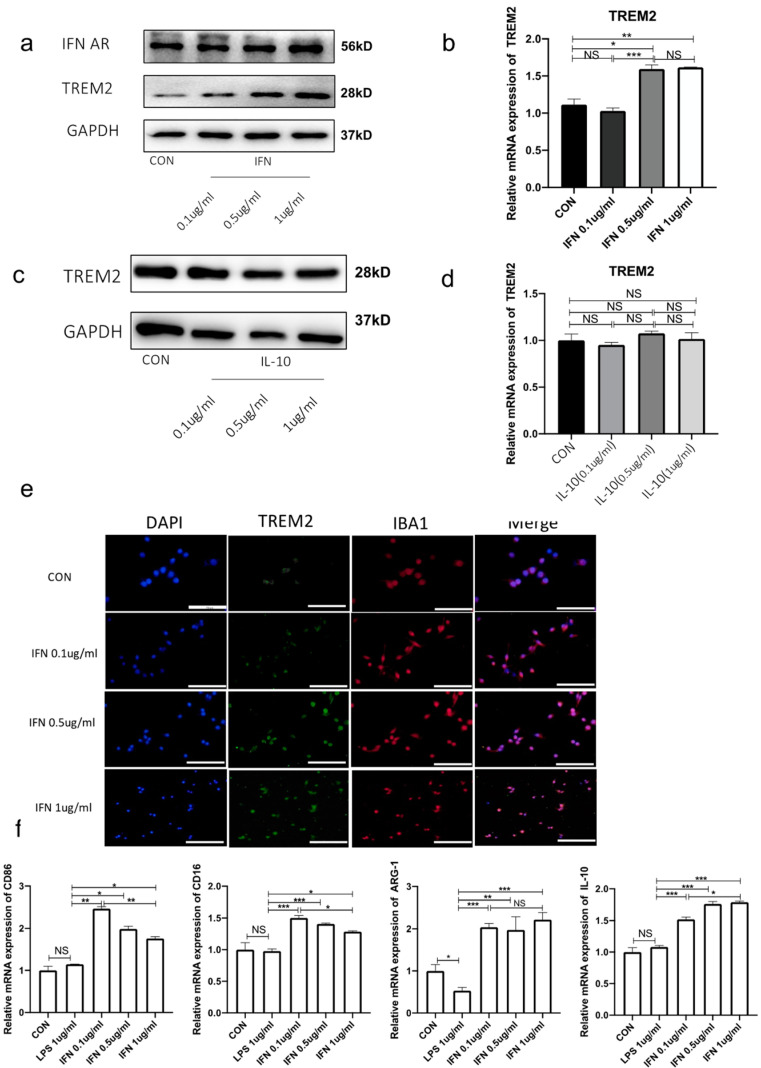
IFN-β increases TREM2 expression and promotes microglia polarization toward M2 phenotype. (**a**) Protein levels of TREM2 after different concentrations of IFN-β. (**b**) mRNA expression of TREM2 after different concentrations of IFN-β by PCR. (**c**) Protein levels of TREM2 after different concentrations of IL-10. (**d**) mRNA expression of TREM2 after different concentrations of IL-10 by PCR. (**e**) Immunofluorescence staining of brain sections from each group. Iba-1 indicates activated microglia (Red). (**f**) mRNA expression of M1-type microglia markers (CD16 and CD86) and M2-type microglia markers (ARG-1 and IL-10) by PCR. NS indicates no statistical difference from the control group. * *p* < 0.05, ** *p* < 0.03, *** *p* < 0.01 compared with control, n = 6 mice per group.

**Figure 7 brainsci-13-00126-f007:**
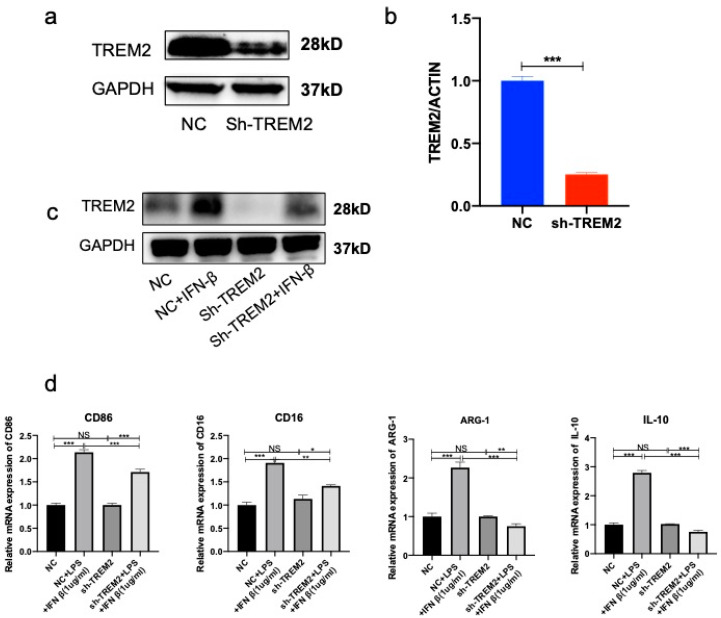
IFN-β increases TREM2 expression and promotes microglia polarization toward M2 phenotype in TREM2 knockdown BV-2 cells. (**a**) Protein expression in shTREM2-transfected BV-2 cells. (**b**) mRNA expression in shTREM2-transfected BV-2 cells. (**c**) Protein expression of TREM2 after 24 h treatment of sh-TREM2 and NC cells with different concentrations of IFN-β. (**d**) mRNA expression of M1-type microglia markers (CD16 and CD86) and M2-type microglia markers (Arg-1 and IL-10) by PCR. NS indicates no statistical difference from the control group. * *p* < 0.05, ** *p* < 0.03, *** *p* < 0.01 compared with control, n = 6 mice per group.

**Table 1 brainsci-13-00126-t001:** Primer sequences used for quantitative PCR.

Gene	Primer Sequences
IL-6	F: CTGCAAGAGACTTCCATCCAG
	R: AGTGGTATAGACAGGTCTGTTGG
IL-4	F: GGTCTCAACCCCCAGCTAGT
	R: GCCGATGATCTCTCTCAAGTGAT
IL-10	F: CTTACTGACTGGCATGAGGATCA
	R: GCAGCTCTAGGAGCATGTGG
TNF-α	F: CAGGCGGTGCCTATGTCTC
	R: CGATCACCCCGAAGTTCAGTAG
TREM2	F: CTGGAACCGTCACCATCACTC
	R: CGAAACTCGATGACTCCTCGG
iNOS	F: GTTCTCAGCCCAACAATACAAGA
	R: GTGGACGGGTCGATGTCAC
CD86	F: TCAATGGGACTGCATATCTGCC
	R: GCCAAAATACTACCAGCTCACT
CD206	F: CTCTGTTCAGCTATTGGACGC
	R: TGGCACTCCCAAACATAATTTGA
CD16	F: AATGCACACTCTGGAAGCCAA
	R: CACTCTGCCTGTCTGCAAAAG
ARG-1	F: CTCCAAGCCAAAGTCCTTAGAG
	R: GGAGCTGTCATTAGGGACATCA
IFN-β	F: AGCTCCAAGAAAGGACGAACA
	R: GCCCTGTAGGTGAGGTTGAT
GAPDH	F: AGGTCGGTGTGAACGGATTTG
	R: GGGGTCGTTGATGGCAACA
β-Actin	F: GTGACGTTGACATCCGTAAAGA
	R: GTGACGTTGACATCCGTAAAGA

## Data Availability

All data reported in this paper will be shared by the corresponding author upon request.

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
