# Peer review of "Consecutive Injection of High-Dose Lipopolysaccharide Modulates Microglia Polarization via TREM2 to Alter Status of Septic Mice"

_brainsci, 2023, doi:10.3390/brainsci13010126_

Round 1

Reviewer 1 Report

This paper has the potential for significant improvement if revised.  I am afraid l found much of it confusing due to the quality of english grammar.  The misuse of verbs and punctuation were numerous.  However, the major problem l have is the overall presentation and interpretation of the experiments.  I found the title of the paper confusing because of all the preliminary descriptions of the experimental model.  However, as the technical side of the experiments appear well done, there is the potential to improve the paper exists.

1)   The concepts presented in this paper about M1 and M2 are out of date and need to be reconsidered.  These designations are not helpful and come from old studies.  Even M2 can be considered as having multiple subgroups.  The authors need to consider the findings in terms of the disease being modeled.

2)  As mentioned the difference between 4 day and 7 day of LPS infection does not make sense without any explanation.  These high doses of LPS undoubtedly made the animals sick as evidenced by loss of weight.  Can the authors state explicitly whether there was a noticeable difference in eating, fever and in particular mortality in the animals between these doses.  I have to add that it is not clearly stated in the text that these injections were administered daily over the period.  I can only conclude this from the figure.

3)  If the authors want to keep the same title to this paper, l would suggest rearranging the figures.  Figures 1 and 2 are really just background controls to their model.  The figure legend to fig 2 is not accurate and should be replaced as none of the measures being presented measure inflammation.  I would suggest moving   Figure 4 to Figure 1 to get to the point of the paper and then present the other data.

4)  The quality of the immunohistochemistry images is not acceptable.  I can not see higher quality images anywhere.  These are crucial for the story being presented and they do not support the paper.  

5)  I always like to see complete images of blots to know that the antibodies are specific.  I notice in supplemental data that the images of complete blots have been cropped as well.  This is particularly important for presenting blot images of TREM2 as it has been well known that many of the commercial antibodies are not specific.  I thought that TREM2 should have a mw of around 44 to represent the glycosylated form, while the 28 kd form represented unglycosylated.  The particular antibody using from CST is qualified for human TREM2 only.  Figure 4 a TREM2 lanes is not convincing and should be improved.  I notice in Figure 4e, there are multiple bands being presented.

6)  This study is missing a control brain region for comparison.  It would be worthwhile to make some of the measurements in cerebellum if possible.  I would be interested if they have any measurements in substantia nigra region as the experimental model is widely used to induce parkinson like syndrome in mice.  

7)  Overall, l am not convinced that the authors have shown that changing TREM2 expression changes polarization.  The changes observed as a result of beta interferon treatment will change activation state and TREM2 expression but the authors have not proven that these changes are dependent on TREM2.  TREM2 has been more clearly defined as a phagocytic marker rather than an activation marker.  The authors should dig more deeply into the literature on this while rewriting their discussion as the reference list is very brief for this topic.

8)  In figure 3, omit the ratio figures.  They are only noise while the raw data figures are the same and more relevant for the reader.

Author Response

非常感谢您的深思熟虑和建设性意见。我们根据您的建议修改了手稿,并在适当的地方添加了新信息。我们很高兴将修改后的手稿重新提交给您,以供进一步考虑。

请参阅附件。

谢谢!

Thank you so much for your thoughtful and constructive comments. We have revised the manuscript based on your suggestions and added new information where appropriate. We are pleased to resubmit the revised manuscript to you for further consideration.

Thanks!

Reviewer 2 Report

Thank you for  a good opportunity. it is necessary for some articles to revise it.

*for introduction

It was able to understand the relations of sepsis ant TREM2 in your manuscript. However you had better to discuss the relationship between sepsis, LPS and TREM2. Furthermore, I was able to understand the importance of this experiment, but you need to explain more detail.

*for line 55th

Is it not necessary to change the line 55th ''On'' of the capital letter to the small letter?

*for material and method

The 148th line word '' chemiluminescence'' is different from other fonts.

It is different from others in a font from the 155th to the 160th line.

*for results

Figure 4 should be divided it into two of a-d and e-h.

All bands of TREM2 are very indistinct in Fig,4e. Which bands do we had better?

You should unify the names of the cell in BV 2 or BV-2.

*for discussion

Are you not thinking about the prospects of the future in this study?

What kind of thing are you thinking this study to be able to develop it in the future? 

Author Response

Thank you very much for  for your thoughtful and constructive comments. We have revised our manuscript according to your suggestions  and have added new information where appropriate. We are pleased to resubmit our revised manuscript to you for further consideration.

Thank you!
